# Anthraquinone Production from Cell and Organ Cultures of *Rubia* Species: An Overview

**DOI:** 10.3390/metabo13010039

**Published:** 2022-12-26

**Authors:** Hosakatte Niranjana Murthy, Kadanthottu Sebastian Joseph, Kee Yoeup Paek, So Young Park

**Affiliations:** 1Department of Botany, Karnatak University, Dharwad 580003, India; 2Department of Horticultural Science, Chungbuk National University, Cheongju 28644, Republic of Korea; 3Department of Life Sciences, CHRIST (Deemed to be University), Bengaluru 560029, India

**Keywords:** anthraquinones, alizarin, bioreactors, cell cultures, dyes, purpurin, secondary metabolites

## Abstract

The *Rubia* genus includes major groups of medicinal plants such as *Rubia cordifolia*, *Rubia tinctorum*, and *Rubia akane*. They contain anthraquinones (AQs), particularly alizarin and purpurin, which have pharmacological effects that are anti-inflammatory, antioxidant, anticancer, hemostatic, antibacterial, and more. Alizarin and purpurin have been utilized as natural dyes for cotton, silk, and wool fabrics since the dawn of time. These substances have been used in the cosmetics and food industries to color products. The amount of AQs in different *Rubia* species is minimal. In order to produce these compounds, researchers have established cell and organ cultures. Investigations have been conducted into numerous chemical and physical parameters that affect the biomass and accumulation of secondary metabolites in a cell, callus, hairy root, and adventitious root suspension cultures. This article offers numerous techniques and approaches used to produce biomass and secondary metabolites from the *Rubia* species. Additionally, it has been emphasized that cells can be grown in bioreactor cultures to produce AQs.

## 1. Introduction

There are more than 60 species in the genus *Rubia* (Family: Rubiaceae), which is found in Asia, Africa, America, and Europe. Among these, *R. cordifolia* is a significant medicinal plant whose roots have been utilized for centuries in Traditional Chinese Medicine (TCM), Traditional Indian Medicine (Ayurveda), and Oriental medicine. It is a blood purifier, immunostimulant, anti-inflammatory, and anti-platelet activating agent used in Ayurveda [1]. This plant has been used in Traditional Chinese medicine (TCM) and oriental medicine to treat illnesses such as heart problems, yellow fever, rheumatism, hematemesis, epistaxis, metrorrhagia, contusion, menoxenia, and arthralgia [2]. The climber and perennial herb *Rubia cordifolia* have long, grooved stems that turn woody at the base (Figure 1A), angular leaves clustered in four whorls (Figure 1B), and tiny flowers placed in terminal panicle cymes (Figure 1C,D). The roots are long, cylindrical, and red in hue (Figure 1E). Pigments are derived from roots. Due to the presence of secondary metabolites in them, several *Rubia* species are well known for their economic and commercial significance. Major secondary metabolites recovered from *Rubia* species include anthraquinones (AQs), naphthoquinones, terpenes, and iridoids; anthraquinones and their glycosides are prominent among them [1,2,3]. Since ancient times, AQs derived from various *Rubia* species have been utilized as natural colorants [4]. The term “Madder” refers to *Rubia* species that produce colorants from different plant parts, particularly roots. Five main species produce AQs and their derivatives: *Rubia tinctorum* L. (also known as Dyer’s madder or European madder), *Rubia peregrina* L. (also known as Wild madder), *Rubia cordifolia* L. (also known as Indian madder), *Rubia sikkimensis* Kurz (also known as Naga madder), and *Rubia yunnanensis* Diels (also known as Xiao hong) [4].

The quinone family contains several compounds known as AQs, which vary in the types and locations of their substituent groups. They are tricyclic aromatic chemical compounds with the formula C_14_H_8_O_2_, and the C-9 and C-10 positions on the central ring are where the ketone groups are located (Figure 2). In general, eight hydrogens can be swapped for each anthraquinone derivative. The term “hydroxyanthraquinoid derivatives” typically refers to 9,10-anthraquinone derivatives in which hydroxyl (-OH) groups have taken the place of numerous hydrogen atoms. The hydroxyanthraquinoid derivatives are pigmented and absorb visible light [5]. AQs are usually accumulated in the roots of *Rubia* species and can exist either in free form or as anthraquinone glycosides [4]. From the *Rubia* species, more than 35 anthraquinone compounds have been discovered; many of these compounds are thought to be artifacts created during extraction or drying [4,6]. Table 1 lists the principal anthraquinone compounds of *R. tinctorum*, *R. peregrina*, *R. cordifolia*, and *R. akane*; Figure 3 shows the structures of a few typical compounds. As natural dyes, these include alizarin (yellow to red), pseudopurpurin (orange), purpurin (dark red), lucidin-3-O-primeveroside (red), ruberythric acid (golden yellow), nordamnacanthal (orange), and munjistin (orange-red) [5]. Fabrics dyed with alizarin and purpurin include cotton, silk, wool, and jute. They are also employed as hair dyes and cosmetics. In Japan and Korea, these substances have been used to color products such as chewing gum, ice cream, and noodles [5].

The biological effects of the anthraquinone derivatives, including their antibacterial, anticancer, anti-platelet aggregation, anti-inflammatory, antioxidant, and anti-hepatitis properties, have been documented [2]. In several in vitro and in vivo tests, the main chemicals alizarin and purpurin, which are isolated from *Rubia* species, have also shown diverse pharmacological activity, including antigenotoxic, anticancer, neuromodulatory, and antimicrobial effects [12,13]. Natural dyes such as alizarin and purpurin have been used to color silk, cotton, wool, nylon, and other textiles. Additionally, they were employed in the production of shampoos, lotions, and sprays, as well as natural hair colors [6].

Di- and tri-hydroxyanthraquinone-glycosides are present in around 2% of the dry weight of the roots of *Rubia* species, particularly *R. tinctorum* and *R. cordifoila*. The presence of anthraquinone molecules, however, may differ depending on the phenology, genotype, and age of plants [14,15]. Plant cell and tissue cultures are attractive alternatives for the production of valuable secondary metabolites, and they are independent of geographical, seasonal, and environmental variations. They offer a defined production system, which ensures a continuous supply of products, uniform quality, and yield. Moreover, cell and tissue cultures are free from contamination by other organisms, pesticides, and insecticides [16]. Many methods can be used to increase the production of secondary metabolites in tissue cultures, including strain enhancement, improvement of the culture conditions and environment, elicitation, feeding of nutrients and precursors, and other bioprocess technologies [16]. These days metabolic engineering techniques have been applied to promote the accumulation of desired compounds in tissue cultures [17]. Scale-up processes are also possible through the application of bioreactor technologies [18,19,20]. Given the foregoing, the *Rubia* species’ anthraquinone production has been adapted through cell and organ cultures. Many methods can be used to increase the production of secondary metabolites in vitro cell and organ cultures, including obtaining efficient cell lines for growth, screening of high-growth cell lines to produce metabolites, manipulation of nutrients to improve yield, optimization of culture environment, elicitation, and organ cultures for production of metabolites. In this review, the potential of employing plant cell and tissue culture methods to increase anthraquinone synthesis is investigated.

## 2. Biosynthesis of Anthraquinones

The structural basis of AQs is an anthracene ring with a basic core of a keto group at positions 9 and 10 and several functional groups at other positions, including -OH, -CH_3_, -OCH_3_, -CH_2_OH, -CHO, -COOH, and others (Figure 2). Plants synthesize AQs and their derivatives as secondary metabolites, and they can exist either as free or as glycosides [5]. When one or more sugar molecules, often glucose or rhamnose, are linked to an aglycone by an O-glycoside bond to a hydroxyl group, glycosides are formed. Some of these compounds are thought to be artifacts created during the extraction or drying of the dyestuff. The roots of *Rubia tinctorum* contain about 35 AQ derivatives [5]. In *Rubia* species, AQs are produced through the chorismite/*o*-succinylbenzoic acid or shikimate pathways (Figure 4). Ring C is created from isopentenyl diphosphate either by the mevalonic acid pathway or the 2-C-methyl-D-erythritol 4-phosphate pathway, whereas rings A and B are produced from chorismite and -ketoglutarate via *o*-succinylbenzoic acid [21,22,23]. The enzyme isochorismate synthase first transforms chorismate, a byproduct of the shikimate pathway, into isochorismate during the biosynthesis of A and B. *o*-succinylbenzoic acid (OSB) synthase, then catalyzes the conversion of isochorismate into OSB in the presence of ketoglutarate and thiamine diphosphate (TPP). Eventually, an OSB-CoA ester is created by activating OSB at the aliphatic carboxyl group. This reaction is catalyzed by OSB: CoA ligase. Ring closure of OSB: CoA produces 1,4-dihydroxy-2-naphtolic acid (DHNA), which gives rise to anthraquinones’ rings A and B. The eventual prenylation of DHNA results in naphthoquinone. The cyclization reaction between naphthoquinone and isoprene unit IPP or 3,3-dimethylallyl diphosphate (DMAPP), a byproduct of the 2-C-methyl-D-erythritol 4 phosphate (MEP) and mevalonic acid (MVA) pathways, results in the synthesis of ring C of anthraquinones [23].

## 3. Callus Cultures for the Production of Anthraquinones

The quantity of AQ accumulation in *Rubia* species is minimal (1.5–2%), and AQ concentrations also vary greatly with plant age, genotype, and phenology [15,21]. As a result, numerous researchers have established callus suspension culturing techniques in *Rubia* species, and the information is presented in Table 2.

### 3.1. Optimization of Chemical and Physical Parameters

On Murashige and Skoog medium [42], Shin [24] established callus cultures of *R. cordifolia* and *R. akane*, and he examined the effects of 2,4-D or NAA (2 ppm), kinetin (5 ppm), temperature (20, 25, 30 °C), and α-ketoglutaric acid (10, 100 mg L*^−^*^1^) or shikimic acid (100 mg L*^−^*^1^) on the growth of callus and the accumulation of metabolites. According to Shin’s [24] findings, NAA promoted callus growth while incubating cultures at 25 °C and supplementing them with α-ketoglutaric acid encouraged the accumulation of alizarin and purpurin in callus cultures of both *R. cordifolia* and *R. akane*. In a different investigation, Mischenko et al. [25] established callus cultures of *R. cordifolia* on MS medium containing 0.5 mg L*^−^*^1^ BA and 2 mg L*^−^*^1^ NAA from the stem, stem apex, leaf, and leaf petiole explants, and selected clones. They were able to raise callus clones/lines from all the explants, but callus clones of stem and leaf petiole origin had the highest purpurin and munjistin contents. They chose clones with yellow and orange coloring and were successful in keeping them alive after extended cultivation.

### 3.2. Transformation of Root Loci (rolA, rolB, rolC) and Other Genes and Elicitation

Bulgakov et al. [26] transformed root loci genes (*rolB* and *rolC* genes) of *Agrobacterium rhizogenes* by cocultivation with *R. cordifolia* leaf and stem segments and raised transformed callus clones. They chose a few transgenic callus clones that outperformed non-transformed clones in terms of biomass accumulation as well as purpurin and munjistin levels. Additionally, they performed elicitation on a few different callus clones using salicylic acid (SA) or methyl jasmonate (MJ), as well as ethephon (0, 1, 10, and 100 μM), and the results revealed that SA and MJ stimulated purpurin and munjistin accumulation 1–1.5 times more than the untransformed clones. They also administered cantharidin, an inhibitor of protein phosphatase 2A, to control and transgenic callus clones, and the results showed that cantharidin supplementation stimulated the induction of AQ in *R. cordifolia* transgenic cells but failed to stimulate any response in the untransformed cultures. The combined action of MJ and cantharidin, which stimulates the accumulation of AQ in transgenic cells, was demonstrated in other studies. Through these tests, they were able to show that cantharidin and MJ work synergistically to change the regulatory balance of secondary metabolism in transformed callus clones of *R. cordifolia*. They also demonstrated that with such treatments, transformed clones had an AQ accumulation of 4% (dry weight) as opposed to non-transformed clones, whose AQ content was 1.5% dry weight. Further, they observed that ethephon (ethylene producer) did not affect AQ synthesis either in the transgenic or non-transformed strains. Moreover, MJ and SA similarly dynamically raised AQ content in both transgenic and non-transformed strains. They concluded that the pathways of ethylene, MJ, and SA are not engaged in the activator function of *rol* genes based on these experimental findings. Bulgakov et al. [27] employed the Ca^2+^ channel blockers verapamil and lanthanum (III) chloride (LaCl_3_) to confirm the involvement of the Ca^2+^-dependent NADPH oxidase signaling pathway in transformed cell lines. Verapamil and LaCl_3_ were used to treat both transformed (*R. cordfiolia*-Rc-*rolC3* and Rc-*rolB*) and untransformed (Rc) clones/lines. They found that there were notable differences between the transformed cultures of *rolC* and *rolB* in terms of their sensitivity to Ca^2+^ channel blockers and calcium deficiency. Compared to the non-transformed culture, the *rolC* culture showed less resistance to the inhibitors, but the *rolB* culture was more resistant to the inhibitors. These findings led them to the conclusion that the Ca^2+-^dependent NADPH oxidase signaling pathway is not activated during the production of AQ in transgenic cultures. Okadaic acid and cantharidin, which are inhibitors of protein phosphatases 1 and 2A, were employed to treat both transformed and untransformed lines in a subsequent experiment to investigate the involvement of various phosphatases in the production of AQ [28]. Their findings demonstrated that cantharidin had no effect, while okadaic acid increased AQ accumulation in both transgenic and non-transformed cultures. According to these findings, AQ synthesis occurs in both normal and transgenic cultures of *R. cordifolia* using various phosphatases.

Using transformed (Rc-*rolC3* and Rc*-rolB*) and non-transformed (Rc) clones/lines, Bulgakov et al. [29] investigated the function of the octadecanoid signaling pathway inhibitors diethyldithiocarbamate, propyl gallate, salicyl hydroxamic acid, and piroxicam. None of the investigated inhibitors of the octadecanoid pathway prevented AQ accumulation in both non-transformed and transformed cultures. Based on these findings, they concluded that the *rolC* and *rolB* gene-mediated increase in AQ synthesis does not include the octadecanoid pathway.

Shkryl et al. [32] transformed the *R. cordifolia* with *rolA*, *rolB*, and *rolC* genes using *Agrobacterium tumefaciens* strain GV3101 harboring constructs pPC002-A (*rolA* under the control of its native promoter), pPCV002-CaMVBT (*rolB* under the control of 35S CaMV promoter), pPCV002-CaMVC (*rolC* under the control of 35S CaMV promoter) and pPCV002-ABC (*rolA*, *rolB*, *rolC* under control of their native promoters) to assess the role of these genes (individually and synergistically) on AQ production in transformed calli. According to the findings of these investigations, the *rolA*, *rolB*, and *rolC* genes alone can increase the synthesis of AQ_S_ in transformed *R. cordifolia* calli. According to Shkryl et al. [32], the isochorismate synthase (ICS) gene, a crucial component of AQ biosynthesis, has higher transcription levels as a result of the stimulatory effect. In contrast to control and non-transformed calli, the transformed calli containing the *rolB* gene displayed vigorous AQ-stimulating activity, resulting in a 15-fold increase in AQ accumulation. Additionally, they demonstrated that a tyrosine phosphatase inhibitor reversed the *rolB*-induced rise in AQ production, demonstrating the role of tyrosine (de)phosphorylation in the stimulation of AQ. When compared to control calli, the *rolA*- and *rolC*-expressing cultures generated 2.8- and 4.3-fold more AQs, respectively. However, their results showed that the effect of *rolA*, *rolB*, and *rolC* on AQ biosynthesis was not synergistic because *rolA* and *rolC* attenuated the stimulatory effect of *rolB* on AQ synthesis.

To ascertain whether *R. cordifolia* cells transformed with the *rolC* gene are engaged in the formation of reactive oxygen species (ROS), which may result in enhanced synthesis of phytoalexins, Bulgakov et al. [31] experimented with *R. cordifolia* cells. They demonstrated lower steady-state levels of ROS in *rolC*-expressing *R. cordifolia* cells compared to control cells using single-cell tests based on confocal microscopy and fluorogenic dyes. While *rolC*-transformed cells were not significantly affected by the ROS inducer paraquat, normal cells saw a considerable increase in ROS. They also demonstrated that *rolC*-transformed cells had a two- to three-fold increase in tolerance to salt, heat, and cold treatments. These findings show that the *rolC* gene does not contribute to the synthesis of phytoalexins and instead functions as a ROS suppressor.

In multiple tests, Shkryl et al. [33] examined the expression of important antioxidant genes in the tissues of *R. cordifolia* leaves as well as in calli produced from leaves that were both non-transformed and pRiA4-transformed. They isolated partial cDNA (complementary DNA) sequences of ascorbate peroxidase, catalase, and Cu/Zn super oxidase dismutase genes (*RcApx1*, *RcApx2*, *RcApx3*, *RcCAT1*, *RcCAT2*, *RcCSD1*, *RcCSD2*, and *RcCSD3*) from plant tissues, as well as pRiA4-transformed and normal calli of *R. cordifolia*, and studied their expression by real-time PCR (polymerase chain reaction). Ascorbate peroxidase (*RcApx1*) and Cu/Zn superoxide dismutase (*RcCSD1*) were found to be the most prevalent transcripts in both plant tissues and untransformed calli, according to transcriptional profiling. In contrast, catalase genes were expressed in such samples. In pRiA4-transformed calli, they noticed the expression of numerous genes encoding ROS-detoxifying enzymes. These findings show that *A. rhizogenes* upregulates its antioxidant genes as a means of lowering ROS levels in the host cells.

In transgenic *R. cordifolia* cells, the effect of heterologous expression of the *Arabidopsis* calcium-dependent protein kinase (CDPK) gene, *AtCPK1*, on the production of AQ was examined by Shkryl et al. [34]. The agrobacterial transformation was employed to introduce a constitutively active, Ca^2+-^independent form and a non-active form (used as a negative control) into *R. cordifolia* cells. They discovered that overexpressing *AtCPK1* in *R. cordifolia* cells resulted in a 10-fold increase in AQ content when compared to untransformed control cells but that *AtCPK1* in the non-active form did not affect AQ production. Shkryl et al. [34] used real-time PCR analysis to demonstrate a correlation between the activation of the isochorismate synthase gene and the inhibition of anthraquinone production in transgenic calli. For up to several years, *AtCPK1*’s activator action remained constant when transgenic cells were grown for extended periods. These findings imply that secondary metabolism in plant cells can be engineered using the CDPK gene. The expression of the *Arabidopsis* CDPK gene, *AtCPK1*, in *R. cordifolia* cells led to a modest and permanent elevation of intracellular reactive oxygen species (ROS) levels, according to a study by Bulgakov et al. [35]. In another study, Bulgakov et al. [35] reported that the expression of the *Arabidopsis* CDPK gene, *AtCPK1*, in *R. cordifolia* cells caused moderate and stable elevation of intracellular reactive oxygen species (ROS) levels. In contrast, no such effect was produced by the mutant, inactive *AtCPK1* gene. Veremeichik et al. [36] investigated the expression of the *R. cordifolia RcPrx01*-*RcPrx07* peroxidase genes in aerial organs, cells transformed with the *rolB* and *rolC* genes, and cells transformed with the wild-type *A. rhizogenes* A4 strain. They demonstrated that all the examined peroxidase genes had significantly higher levels of expression in *rolB*-overexpressing cells than in other cells. These findings imply that the *R. cordifolia* native genes are overexpressed as a result of the agrobacterial *rolB* gene. Bulgakov et al. [37] showed that the *rolB* gene inhibits the production of reactive oxygen species (ROS) induced by paraquat, menadione, and light exposure in *R. cordifolia* cells that express this gene. These findings support the role of the *rolB* genes in the metabolism of ROS in transformed cells.

In order to measure the expression level and investigate their effects on AQ production, Shkryl et al. [39] generated native and constitutively active (Ca^2+^-independent) versions of *AtCPK1* in *Rubia cordifolia* cells. In *AtCPK1* lines, they measured the expression of genes encoding crucial AQ biosynthesis pathway enzymes such as isochorismate synthase (ICS), *o*-succinylbenzoate synthase (OSBS), *o*-succinyl benzoate ligase (OSBL), and isopentenyl diphosphate isomerase (IPPi). They reported enhanced expression of the ICS, OSBS, OSBL, and IPPI genes, as well as higher AQ synthesis in all *AtCPK1*-transgenic cell lines. These findings support the function of the *R. cordifolia* ICS, OSBS, OSBL, and IPPI genes in AQ biosynthesis. In a further investigation, Veremeichik [40] showed that the callus lines that were retained throughout long-term cultures expressed the *rolA* gene at a high level consistently.

### 3.3. Immobilization and Other Strategies

A method for immobilizing *Rubia tinctorum*-suspended cells was developed by Nartop et al. [38] using lignocellulosic material from jute, sisal, and loofa sponge. In comparison to control cell suspension, the immobilized cells produced 6.05 and 22.91 times more alizarin and purpurin when cultivated in a nutrient medium. In a different study, Mariadoss et al. [41] exposed the *R. cordifolia* callus to 2, 4, 6, 8, 10, 12, 14, and 16 Grays of gamma irradiation. The gamma-irradiated cells were then cultured in an MS medium that contained 1 mg L*^−^*^1^ of IAA, 1 mg L*^−^*^1^ of NAA, and 1 mg L*^−^*^1^ of BA, and growth and metabolite accumulation were evaluated. According to Mariadoss et al. [41], the callus cultures that underwent gamma irradiation at eight Grays accumulated a maximum alizarin level of 26.86 mg g*^−^*^1^ DW and a purpurin level of 44.85 mg g*^−^*^1^ DW during the fourth sub-cultures. These findings support the notion that mutant cells can accumulate more secondary metabolites. The presence of munjistin, purpurin, ruberythrinic acid, alizarine, xanthopurpurin, and munjistin methyl ester was reported by Mishchenko et al. [30] who carried out the qualitative and quantitative assessment of AQ in *Rubia cordifolia* cell cultures. In a model of edema, they also showed that the cell culture extract of *R. cordifolia* had anti-inflammatory properties.

## 4. Cell Suspension Cultures for the Production of Anthraquinones

The research conducted on cell suspension cultures in several *Rubia* species is shown in Table 3. Suzuki and colleagues [43] established cell suspension cultures in *Rubia cordifolia* and investigated the growth kinetics and effects of sucrose concentrations (2–7%), several sugars (sucrose, fructose, glucose, raffinose, lactose, and rhamnose at 5% level), myo-inositol concentrations (10–250 mg L*^−^*^1^), inorganic ions (KNO_3_, 0.1–1.0%), nitrogen source (NH_4_ or NO_3_ or combination of NH_4_:NO_3_, 1:3–3:1), auxins (NAA, IAA, 2,4-D; 0.2–1.0 mg L*^−^*^1^), and NAA (0.2–5.0 mg L*^−^*^1^) on cell growth and AQ content. According to their findings, there was a normal lag phase lasting 4 days, an exponential period lasting 20 days, and a stable phase with the accumulation of cell biomass and AQs after 20 days. The combination of 20 mg L*^−^*^1^ myoinositol, 5% sucrose, a 1:1 NH_4_:NO_3_ ratio, and 0.4 mg L*^−^*^1^ NAA was determined to be the most effective for biomass accumulation and AQ production among the numerous parameters examined. The effects of pH, white, blue, and red fluorescent light conditions on the development of cultivated cells and the synthesis of AQ were examined by Suzuki et al. [44]. They reported pH changes had not affected the growth and accumulation of AQs. Moreover, the incubation of cultures under dark conditions showed the highest accumulation of biomass and metabolites rather than monochromatic red, blue light, or fluorescent light conditions. On MS medium supplemented with 5 μM NAA+0.1 μM KN and MS +0.5 μM 2,4-D, respectively, Sato et al. [45] produced callus from leaf segments of *R. tinctorum* and *R. akane*. They used LS medium containing 0.5 μM NAA + 0.1 μM KN for 20 days in the dark to establish cell suspension cultures in 10 L fermenters. Multiple AQ were found in the cell suspension culture extracts, according to their findings. In elicitation experiments with cell cultures of *R. akane*, Jin et al. [46] investigated the effects of various polysaccharides, including chitosan, alginate, carrageenan, yeast extract, gum arabic, lichenan, xylan, and nigeran, at concentrations of 20–60 mg L^–1^. In the presence of 25 mg L^–1^ chitosan, the total production of AQ increased 1.3 times in an MS medium containing galactose. *Pythium aphanidermatum* (fungus) extract was employed by van Tegelen [47] to generate the *R. tinctorum* cell culture and as an elicitor, which increased AQ production by two-fold. They were able to isolate isochorismate synthase (EC 5.4.99.6) isoforms, the essential enzyme in the production of AQ, using these induced cells.

Eichinger et al. [48] established cell cultures of *R. tinctorum* using radiolabeled (1-^13^C)- or (U-^13^C_6_) glucose as a supplement. These investigations helped understand the AQ metabolic route because they used ^13^C labeling patterns to reconstitute the labeling patterns of acetyl CoA, pyruvate, phosphoenol pyruvate, erythrose 4-phosphate, and α-ketoglutarate through retrosynthesis.

The *R. tinctorum* cell culture system was employed by Vasconsuelo et al. [49,50,51,52] to examine signal transduction pathways. They established cell cultures of *R. tinctorum* by using B5 medium + 2% sucrose + 2 mg L*^−^*^1^ 2,4-D + 0.5 mg L*^−^*^1^ NAA, 0.5 mg L*^−^*^1^ IAA, and 0.5 mg L*^−^*^1^ KN and elicit the cultures using 200 mg L*^−^*^1^ chitosan. They found that chitosan enhanced AQ production in *R. tinctorum* cell cultures via activating the Ca^2+^ messenger, phospholipase C, protein kinase C, phosphoinositide 3-kinase (PI3K), and mitogen-activated protein kinase (MAPK) pathways.

In cell suspensions of *R. tintorium*, Orban et al. [53] investigated the effects of various elicitors, including JA, SA, and polysaccharides of fungal origin, using MS media supplemented with 3% sucrose +1 mg L*^−^*^1^ IAA, 0.2 mg L*^−^*^1^ NAA, and 0.2 mg L*^−^*^1^ KN. They observed a three-fold increase in AQ accumulation, particularly in lucidin primeveroside and ruberythric acid, in the stimulated cells. In cell suspensions of *R. tinctorum*, Perassolo et al. [54] tested the impact of proline and aminoindan-2-phosphonic acid on AQ production and demonstrated that both substances enhanced AQ production. In a different investigation, Perassolo et al. [55] investigated the effects of glutamate and several proline analogs on the pentose phosphate pathway (PPP), the proline cycle, and AQ synthesis in *R. tinctorum* cell suspension cultures. According to these findings, PPP is not a limiting factor as a carbon donor to the shikimate pathway or for the synthesis of AQs because the treatments have not resulted in the induction of PPP.

## 5. Hairy and Adventitious Root Cultures for the Production of Anthraquinones

Alternatives to cell suspension cultures include cultures of hairy and adventitious roots because they are distinct organs and have a greater ability to participate in primary and secondary metabolism. In order to produce secondary metabolites, hairy and adventitious root cultures have been induced in numerous plants [59,60,61]. For the production of anthraquinones in *Rubia* species, hairy and adventitious root cultures were developed by several researchers. Such reports are compiled in Table 4. In *Rubia cordifolia* var. *pratensis*, Shin and Kim [62] produced hairy roots from stem segments by co-cultivating the plant with *Agrobacterium rhizogenes* A4 strain 15,834. In Nitsch and Nitsch’s medium enriched with 0.5 mg L*^−^*^1^ NAA, hairy root suspension cultures were developed, and such cultures were capable of accumulating AQs. To transform *R. peregrina*, Lodhi and Charlwood [63] co-cultivated callus cultures with *Agrobacterium rhizogenes* LBA 9402 and produced hairy roots. On Gamborg B5 medium with 30 g L*^−^*^1^ sucrose, they later cultivated hairy roots. In comparison to field-grown roots, they discovered a 2-fold increase in AQ accumulation. In a different experiment, Lodhi et al. [64] examined the expression of bacterial isochorismate synthase (EC 5.4.99.6) in transgenic hairy root cultures of *R. peregrina*. They found that after 10 days in culture, transgenic roots containing bacterial isochorismate synthase cDNA expressed twice as much isochorismate synthase activity (4.88 pkat/mg protein) as the control roots (2.45 pkat/mg protein). AQ levels accumulated to 20% after 30 days in culture.

By co-cultivating *Agrobacterium rhizogens* R1000, Park et al. [65] successfully induced hairy roots in *R. akane* and established hairy root cultures in MS liquid media for 25 days. They were able to produce 3.9 mg g*^−^*^1^ DW alizarin and 4.5 mg g*^−^*^1^ DW purpurin through such experiments. In a second experiment, Park and Lee [66] investigated the effects of various concentrations of IAA, IBA, or NAA (0, 0.1, 0.5, and 1 mg L*^−^*^1^) added to B5, MS, and SH media. Their findings demonstrated that hairy roots produced the maximum levels of alizarin (5.9 mg g*^−^*^1^ DW) and purpurin (7.2 mg g*^−^*^1^ DW) production when cultivated in full-strength SH medium with 0.5 mg L*^−^*^1^ NAA.

Sato et al. [67] generated hairy roots in *R. tinctorum*, which were then grown in an MS medium containing 3% sucrose. They investigated the effects of phytohormones 0.5 and 5.0 μM KN, IAA, NAA, or 2,4-D and found that 5 μM NAA was beneficial for AQ production. Additionally, they examined the effects of sucrose concentrations of 6, 8, 12, 15, and 18%, finding that 12% was the best concentration for AQ accumulation. Kino-oka et al. [68] tested the effect of fructose, galactose, lactose, maltose, and sucrose as well as nitrate (KNO_3_) or ammonium form of nitrogen (NH_4_NO_3_) added to MS medium on the synthesis of AQ. According to their findings, fructose was a superior carbon source, and nitrate from nitrogen was beneficial for AQ accumulation. With the hairy root cultures of *R. tinctorum*, Perassolo et al. [69] conducted elicitation tests. Hairy roots were cultivated in 1/2 strength B5 medium with 2% sucrose. They used 100 μM methyl jasmonate to stimulate the hairy root cultures. In comparison to control cells, they detected a 2.4-fold increase in intracellular and an 8.1-fold increase in extracellular AQ hyperaccumulation.

In *Rubia tinctorum*, Bicer et al. [70] developed adventitious roots from internode explants on an MS medium containing 15 μM IBA and 0.5 μM KN. The roots were grown in an MS liquid medium containing 3% sucrose, 1, 2, and 10 and 100 μM methyl jasmonate, as well as 1 and 2 mM caffeic acid. Among the treatments, 2 mM caffeic acid plus 100 mM methyl jasmonate helped AQs accumulate more effectively. Salicylic acid (20 and 40 μM) and L-phenylalanine (50 and 100 μM) were investigated by Demirci et al. [71] for their effects on the accumulation of AQs in adventitious root cultures of *R. tinctorum*. Their findings showed that salicylic acid considerably enhanced AQ accumulation while L-phenylalanine had no discernible effect on AQ accumulation. A 20 μM salicylic acid treatment led to the accumulation of 31.47 mg g*^−^*^1^ DW AQ, which was 1.5 times more than the control. The aforementioned findings demonstrate that numerous *Rubia* species can produce AQs using both hairy root and adventitious root cultures.

## 6. Bioreactor Cultures for the Production of Anthraquinones

The bioreactor culture system offers more benefits than the conventional tissue culture system because the various factors such as aeration, gases such as oxygen, carbon dioxide, and ethylene levels, and hydrogen ion concentration could be regulated in the bioreactors. Continuous medium agitation can maximize nutrient concentration and also improve nutrient absorption. Enhancing cell proliferation and regeneration rates can also speed up production, decrease costs, improve product quality, eliminate pesticide contamination, and allow for year-round harvesting to satisfy the growing demand on a worldwide scale [17,73]. In plants, including Korean ginseng, Siberian ginseng, echinacea, and St. Jones wort, bioreactor culture systems have been effectively developed for the synthesis of important secondary metabolites [74,75,76,77]. Researchers conducted bioreactor experiments on the plant *Rubia tinctorum*, and data from those studies are shown in Table 5. *R. tinctorum* cells were cultivated in 12 and 24-L airlift fermenters using MS media by Laszlo et al. [78], who also produced AQs such as ruberythric acid, alizarin, and purpurin. Busto et al. [79] studied the hydrodynamic stress on biomass and AQ production while growing *R. tinctorum* cells in 1.5 L stirred tank bioreactors at 450 rpm and in Erlenmeyer flasks at 100 rpm. They discovered that although the biomass produced in the bioreactor was 29% less than that produced in the Erlenmeyer flasks, the production of hydrogen peroxide in the bioreactor was 15 times higher, which resulted in a 233% increase in the production of AQ. In a different investigation, Busto et al. [80] compared normal flasks and baffled shaking flasks and established the impact of light irradiation on the generation of biomass and AQ. The generation of AQ is significantly inhibited by light, and dark incubation has been determined to be ideal for *R. tinctorum* cell cultures. After gamma irradiation, Mariadoss et al. [41] chose superior *R. tinctorum* cell lines and cultured them in 8 L stirred bioreactors with different impellers, including helical ribbon impellers (length: 180 mm, width: 90 mm, shaft height: 360 mm), and Rushton turbine impellers (length: 200 mm, width: 50 mm), using MS medium supplemented with 1 mg L*^−^*^1^ BA + 1 mg L*^−^*^1^ NAA+ 1mg L*^−^*^1^ IAA. They discovered that, after 30 days, helical ribbon agitation at a speed of 60 rpm produced a uniform bulk flow of the suspension cultures in the bioreactor without causing shear damage to the cells and produced a maximum biomass of 32.53 g L*^−^*^1^ dry biomass. They also found that using Rushton turbine impeller cells produced 21.04 mg g*^−^*^1^ DW alizarin and 51.28 mg g*^−^*^1^ DW purpurin, but using helical ribbon impeller cultured cells produced 37.96 mg g*^−^*^1^ DW alizarin and 78.93 mg g*^−^*^1^ DW purpurin. These findings point to how crucial it is to choose the right cell lines and to continue to optimize the impellers and agitation speed of cell suspensions to produce bioactive compounds.

## 7. Conclusions and Prospects

Anthraquinones (AQs), which are utilized as natural colors in the food, cosmetic, and textile industries, are produced by *Rubia* species. Additionally, some AQs have a variety of biological functions, which makes them useful to the pharmaceutical industry. For the production of AQs, scientists have developed cell and organ cultures, and other methods for the accumulation of these substances have also been devised. Few researchers have used hairy root and adventitious cultures; the majority of investigations have focused on callus and cell cultures. Research on *R. cordifolia*, *R. tinctorum*, and *R. peregrina* should focus on the induction of adventitious roots and hairy roots, the selection of superior clones, the optimization of the culture media, and physical parameters that affect the accumulation of biomass and AQs. It is important to standardize the cultivation of adventitious and hairy roots in bioreactors as well as the optimization of bioreactor bioprocess parameters. A detailed investigation of the biosynthesis pathway elucidation is urgently required to identify the critical enzymes and genes regulating their expression. As a result, a metabolic route for AQ chemicals produced specifically will be controlled and modified.

## Figures and Tables

**Figure 1 metabolites-13-00039-f001:**
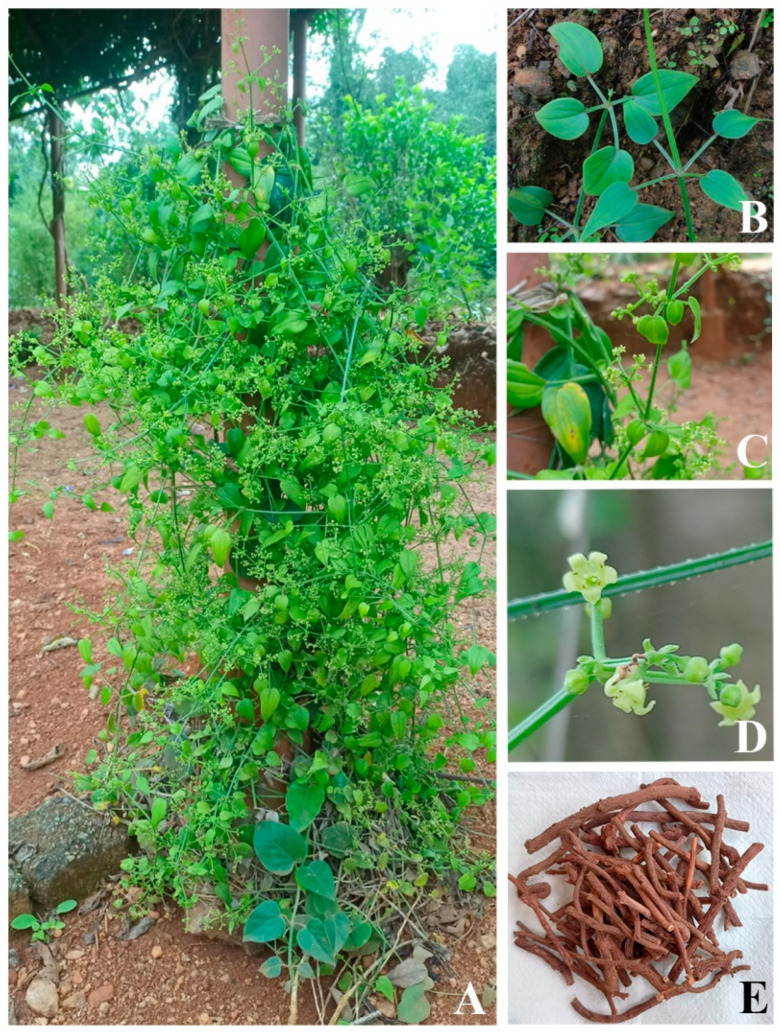
*Rubia cordifolia* plant (**A**), leaves (**B**), flowering twigs (**C**,**D**), dried roots (**E**) (photos were taken by the authors).

**Figure 2 metabolites-13-00039-f002:**
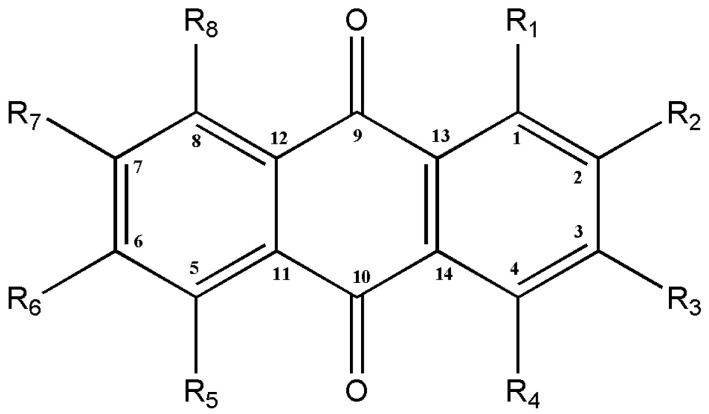
The general structure of anthraquinones.

**Figure 3 metabolites-13-00039-f003:**
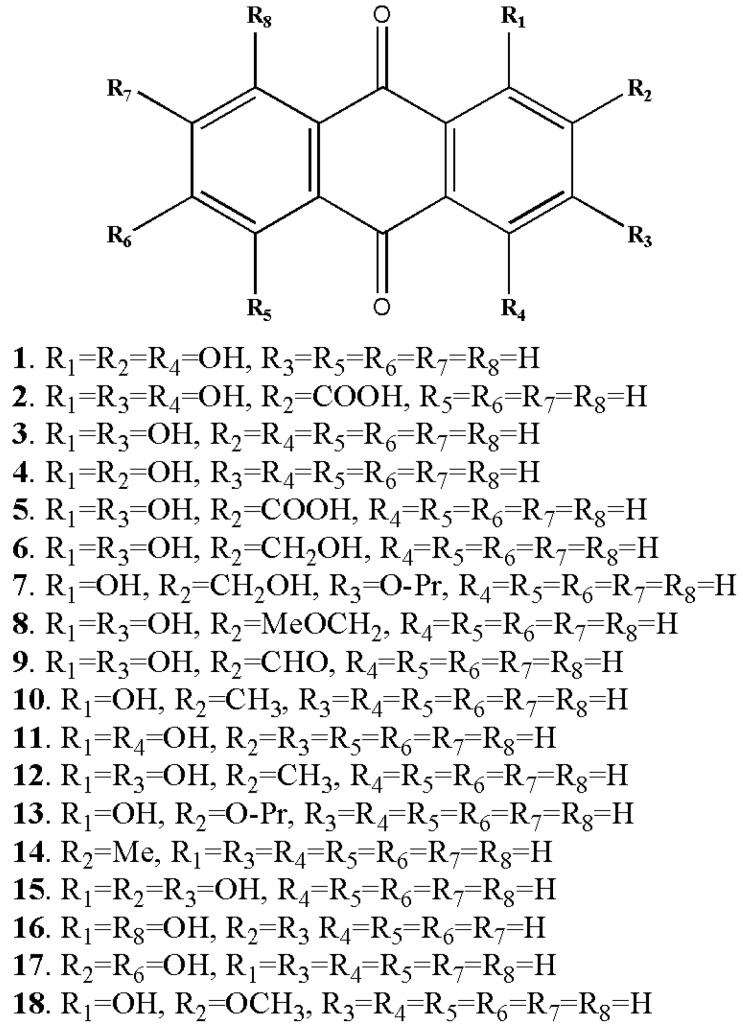
Some of the anthraquinone derivatives isolated from *Rubia cordifolia*. Purpurin (1), pseudopurpurin (2), xanthopurpurin (3), alizarin (4), munjistin (5), lucidin (6), lucidin primeveroside (7), lucidin-Ω-ethyl-ether (8), nordamnacanthal (9), pachybasin (10), quinzarin (11), rubiadin (12), ruberythric acid (13), tectoquinone (14), anthragallol (15), danthron (16), anthrafalvin (17), alizarin-2-methyl-ether (18) (structures of metabolites were drawn by authors).

**Figure 4 metabolites-13-00039-f004:**
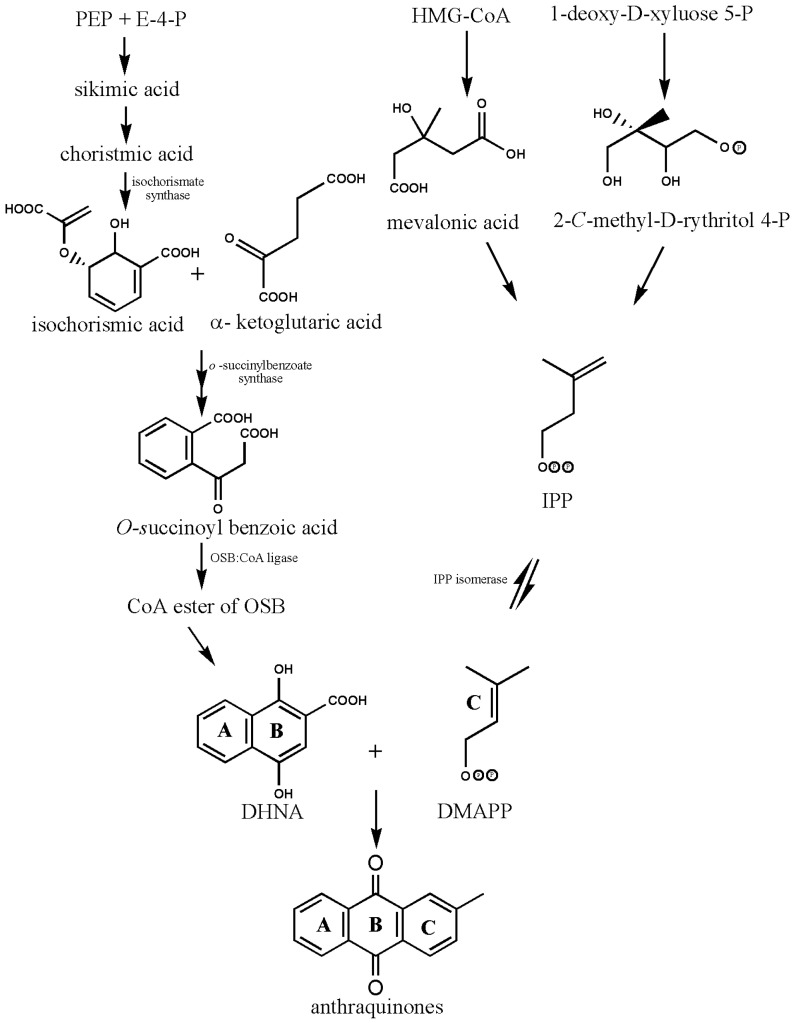
Biosynthesis of anthraquinones. PEP—Phosphoenolpyruvate; E-4-P—Erythrose-4-phosphate; OSB—*o*-succinylbenzoic acid, DHNA—1,4-dihydroxy-2-naphthoic acid, HMG-CoA—3-hydroxy-3-methylglutaryl coenzyme A, IPP—isopentenyl diphosphate, DMAPP—3,3-dimethylallyl diphosphate (biosynthetic pathway and structures were drawn by authors).

**Table 1 metabolites-13-00039-t001:** Naturally occurring anthraquinonoid pigments in *Rubia* species.

Species	Common Name	Major Anthraquinone Compounds	References
*R. peregrina* L.	Wild madder	Pseudopurpurin	[6]
*Rubia tinctorum* L.	Dyer’s madder or European madder	Alizarin, alizarin-2-methyl ether, anthraflavin, anthragallol, danthron, lucidin, lucidin perimerveroside, lucidin, lucidin-ω-ethyl-ether, munjistin, munjistin ethyl ether, nordamnacanthal, purpurin, pseudopurpurin, quinizarin, ruberythric acid, xanthopurpurin	[7,8,9]
*R. cordifolia* L.	Indian madder	Alizarin, lucidin, munjinstin, pseudopurpurin, purpurin, rubiadin, tectoquinone, xnathopurpurin	[6,9,10]
*R. akane* Makino	Japanese madder	Purpurin, ruberythric acid	[6,11]

**Table 2 metabolites-13-00039-t002:** Callus cultures for the production of anthraquinones (AQs) in *Rubia* species.

Species	Medium, Growth Regulators and Parameters Studied	Metabolites and Yield	References
*Rubia cordifolia* and *R. akane*	MS medium with kinetin (5 ppm) + 2,4-D or NAA (2 ppm)	AQs—Not reported	[24]
*Rubia cordifolia*	MS medium 0.5 mg L^−1^ BA and 2 mg L^−1^ NAA	AQs—0.62–1.22% DW	[25]
*Rubia cordifolia*	Transformed with 35S-*rolB* and 35S-*rolC* genes and compared transgenic callus with non-transformed callus cultures; additionally, callus cultures were treated with (1, 10, 100 μM) MJ or SA or ethephone	AQs—4.33% DW	[26]
*Rubia cordifolia*	Studied the effect of Ca^2+^ channel blockers (verapamil and L-type Ca^2+^) and inhibitors of protein phosphatases (protein phosphatase 1 and 2A) on growth and anthraquinone production in non-transformed and *rolB* and *rolC* transformed callus	AQs—3.02% DW	[27]
*Rubia cordifolia*	Studied the effect of *rol* genes on AQ content and showed that induction of AQ production in transgenic cultures does not involve the activation of Ca^2+^-dependent NADPH oxidase pathway	AQs—3.17% DW	[28]
*Rubia cordifolia*	Studied the effect of octadecanoid pathway inhibitors (diethyldithiocarbamate, propyl gallate, salicylhydroxamic acid, and piroxicam) on the synthesis of AQs	AQs—5.00% DW	[29]
*Rubia cordifolia*	Analyzed the chemical composition and pharmacological activity of AQs obtained from callus cultures	AQs—2.14%	[30]
*Rubia cordifolia*	Studied the effect of the *rol*C gene on the production of reactive oxygen species and AQ	AQs—Not reported	[31]
*Rubia cordifolia*	Studied the effect of *rolA*, *rolB*, and *rolC* genes individually and combined the effect of the above genes on the accumulation of AQs	AQs—5.85% DW	[32]
*Rubia cordifolia*	Studied the expression of the key antioxidant gene in *R. cordifolia* transformed with *Agrobacterium rhizogenes* pRiA4-transformed calli	AQs—Not reported	[33]
*Rubia cordifolia*	Transformed calcium-dependent protein kinase gene of *Arabidopsis* (*AtCPK1*) into *R. cordifolia*	AQs—4.36% DW	[34]
*Rubia cordifolia*	Studied the effect of *AtCPK1* transgenic *R. cordifolia* cells on the production of reactive oxygen species	AQs—Not reported	[35]
*Rubia cordifolia*	Studied the expression of peroxidase genes that were isolated from *R. cordifolia,* namely *RcPrx01*-*RcPrx07*, in *R. cordifolia* in aerial organs as well as in cells transformed with the *rolB* and *rolC* genes	AQs—Not reported	[36]
*Rubia cordifolia*	Studied the effect of *rolB* gene in the suppression of reactive oxygen species in *R. cordifolia rolB*-expressing cells.	AQs—Not reported	[37]
*Rubia cordifolia*	Immobilization of suspension cultures with loofa sponge, sisal, and jute mixture (2 g each) and re-cultured in MS liquid medium	AQs—1.32 and 0.55 mg g^−1^ DW	[38]
*Rubia cordifolia*	Transformation with the *AtCPK1* gene (calcium-dependent protein kinase) and obtained increased AQs accumulation	AQs—4.64% DW	[39]
*Rubia cordifolia*	Studied the effect of *rolA* gene expression in long-term cultured callus cultures and obtained increased AQs accumulation	AQs—45 mg g^−1^ DW	[40]
*Rubia cordifolia*	Studied the effect of 2, 4, 6, 8, 10, 12, 14, and 16 Gy of gamma irradiation and callus was cultured on MS + 1 mg L^−1^ IAA + 1 mg L^−1^ NAA + 1 mg L^−1^ BA and the callus cultures irradiated at 8 Gy showed optimum accumulation of AQs	AQs—26.86 and 44.85 mg g^−1^ DW	[41]

^1.^*AtCPK*—*Arabidopsis thaliana* calcium-dependent protein kinase gene; BA—benzyl adenine; 2,4-D—2,4-dichlorophenoxyacetic acid; IAA—indoleacetic acid; KN—kinetin; MJ—Methyl jasmonate; MS—Murashige and Skoog medium [42]; NAA—1-naphthalene acetic acid; SA—salicylic acid; *rolABCD* genes—root loci A, B, C and D genes of root inducing (Ri) plasmid.

**Table 3 metabolites-13-00039-t003:** Cell suspension cultures for the production of AQs in *Rubia* species.

Species	Culture Medium, Growth Regulators, Additives, and Other Parameters	Metabolites and Yield or Productivity	References
*Rubia cordifolia*	MS medium and studied the effect of auxins IAA or NAA or 2,4-D; concentrations of NAA (0.2–5.0 mg L^−1^); ratio of NH_4_ nitrogen and NO_3_ nitrogen (only NO_3_, 1:1, 2:1, 3:1, only NH_4_); sucrose, fructose, glucose, galactose, raffinose, lactose, rhamonose; sucrose concentration (2–7%), myo-inositol concentration (10–250 mg L^−1^) on the synthesis of anthraquinones	AQs—50 μmol g^−1^ day^−1^	[43]
*Rubia cordifolia*	MS + 2.0 mg L^−1^ NAA + 0.2 mg L^−1^ KN +sucrose (5%) +10 mg L^−1^ myo-inositol and studied effect of light on AQ production	AQs—50 μmol g^−1^ day^−1^	[44]
*Rubia tinctorum* and *R. akane*	Callus induction: MS + 5 μM NAA + 0.1 μM KN for *R. tinctorum*; MS + 0.5 μM 2,4-D for *R. akane*. Cell suspension cultures were established by using LS +0.5 μM NAA + 0.1 μM KN	AQs—Not reported	[45]
*Rubia akane*	SH + 2 mg L^−1^ 2,4-D + 3% sucrose; Various elicitors chitosan, alginate, kappa carrageenan, yeast extract, gum Arabic, lichenan, xylan, nigeran were tested at 0, 20, 40, and 60 mg L^−1^.	AQs—320 mg g^−1^ DW	[46]
*Rubia tinctorum*	B5 medium + 2% sucrose + 2 mg L^−1^ 2,4-D + 0.5 mg L^−1^ NAA, 0.5 mg l^−1^ IAA, and 0.5 mg L^−1^ KN and cells were elicited with an autoclave extract of *Pythium aphanidermatum*	AQs—Not reported	[47]
*Rubia tinctorum*	B5 medium + 10 μM 2,4-D + 1.5% (1-^13^C)- or (U-^13^C_6_) glucose and studied the ^13^C labeling pattern of CoA, pyruvate, phosphoenol pyruvate, and others to study the biosynthesis of anthraquinones	AQs—Not reported	[48]
*Rubia tinctorum*	B5 medium + 2% sucrose + 2 mg L^−1^ 2,4-D + 0.5 mg L^−1^ NAA, 0.5 mg L^−1^ IAA and 0.5 mg L^−1^ KN and cells were elicited 200 mg L^−1^ chitosan. They studied signal transduction pathways	AQs—Not reported	[49,50,51,52]
*Rubia tinctorum*	MS + 3% sucrose +1 mg L^−1^ IAA + 0.2 mg L^−1^ NAA +0.2 mg L^−1^ KN; Elicitors 3, 7, 15, and 30 μL mL^−1^ JA or 13, 27, 67.5 and 100 μL mL^−1^ SA or 80 mg mL^−1^ polysaccharides isolated from fungus *Coriolus versicolor*	AQs—70 to 262 mg g^−1^ DW	[53]
*Rubia tinctorum*	B5 medium + 2% sucrose + 2 mg L^−1^ 2,4-D + 0.5 mg L^−1^ NAA, 0.5 mg L^−1^ IAA and 0.5 mg L^−1^ KN; Effect of addition of 0.25 mM proline or 100 μM aminoinda-2-phosphonic acid on anthraquinone production	AQs—1.5 μmol g^−1^ FW	[54]
*Rubia tinctorum*	B5 medium + 2% sucrose + 2 mg L^−1^ 2,4-D + 0.5 mg L^−1^ NAA, 0.5 mg L^−1^ IAA, and 0.5 mg L^−1^ KN; Studied the effect of the addition of glutamate and proline analogs (azatidine-2-carboxylic acid and thiazolidine-4-carboxylic acid) on the pentose phosphate pathway, the proline cycle and anthraquinone production	AQs—Not reported	[55]

^1.^ B5—Gamborg B5 medium [56]; BA—benzyl adenine; 2,4-D—2,4-dichlorophenoxyacetic acid; IAA— indoleacetic acid; JA—Jasmonic acid; KN—kinetin; LS—Linsmaier and Skoog medium [57]; MJ—Methyl jasmonate; MS—Murashige and Skoog medium; NAA—1-naphthalene acetic acid; SA—salicylic acid; SH—Schenk and Hildebrandt medium [58].

**Table 4 metabolites-13-00039-t004:** Hairy/adventitious root cultures for the production of AQs in *Rubia* species.

Species and Type of Root Cultures	Culture Medium	Growth Regulators, Additives, and Other Parameters	Metabolites and Yield or Productivity	References
*Rubia cordifolia* var. *pratensis*; hairy roots	NN	They tested the effect of auxins viz. 0.5, 1.0 mg L^−1^ NAA, and 0.1 mg L^−1^ IAA on the growth and accumulation of metabolites.	AQs—Not reported	[62]
*Rubia peregrina*; hairy roots	B5	Gamborg B5 medium containing 30 g L−1 sucrose, 10 g L^−1^ agar	AQs—2.12 mg g^−1^ DW	[63,64]
*Rubia akane*; hairy roots	MS	MS + 3% sucrose studied the accumulation of alizarin and purpurin	AQs—3.9 and 4.5 mg g^−1^ DW alizarin and purpurin	[65]
*Rubia akane*; hairy roots	Various media	Studied the effect of half and full strength of B5, half and full strength of MS, and a half and full strength of SH media; 0.1, 0.5, 1.0 mg L^−1^ IAA, IBA and NAA	AQs—5.9 and 7.2 mg g^−1^ DW alizarin and purpurin	[66]
*R. tinctorum;* hairy roots	MS	Studied the effect of 0.5, 5 μM IAA, NAA or 2,4-D, and 0.5 μM KN; and 3,6, 9, 12, 15, and 18% sucrose on the growth of roots and AQs production	AQs—700 μg g^−1^ FW	[67]
*Rubia tinctorum*; hairy roots	MS	Studied the effect of fructose galactose, glucose, lactose, maltose, and sucrose (2%); altered nitrogen (1.90 g L^−1^ KNO_3_ + 1.65 g L^−1^ NH_4_NO_3_) and (6.07 g L^−1^ KNO_3_) on AQs accumulation	AQs—0.72 mg g^−1^ day^−1^	[68]
*Rubia tinctorum*; hairy roots;	1/2 B5	2% sucrose + 2 mg L^−1^ 2,4-D + 0.5 mg L^−1^ NAA, 0.5 mg L^−1^ IAA and 0.5 mg L^−1^ KN and cells were elicited by 100 μM MJ	AQs—33.4 μmol L−1 day^−1^	[69]
*Rubia tinctorum*; adventitious roots	MS	Studied the effect of Methyl jasmonate (10 and 100 μM) and caffeic acid (1 and 2 mM) on the growth of roots and AQs	AQs—Not reported	[70]
*Rubia tinctorum;* adventitious roots	MS	MS +2% sucrose and studied the effect of L-phenylalanine (50 and 100 μM) and salicylic acid (20 and 40 μM) on the accumulation of AQs	AQs—31.47 mg g^−1^ DW	[71]

^1^ 2,4-D—2,4-dichlorophenoxyacetic acid; Gamborg B5 medium; IAA—indoleacetic acid; KN—kinetin; MJ—Methyl jasmonate; MS—Murashige and Skoog medium; NAA—1-naphthalene acetic acid; Nitsch and Nitsch medium [72]; SA—salicylic acid.

**Table 5 metabolites-13-00039-t005:** Production of AQs in *Rubia* species using bioreactor cultures.

The Type of Bioreactor Used	Species and Type of Culture	Culture Medium and Growth Regulators/Additives	Optimization of Factors	Metabolites and Yield or Productivity	References
New Brunswick fermenters; 1.5 L	*Rubia cordifolia*, cell suspension culture	MS medium + 1 mg L^−1^ IAA + 1 mg L^−1^ NAA + 1 mg L^−1^ BA and 3% sucrose	The effect of helical and Rushton turbine impellers on the accumulation of anthraquinones	AQs—37.96 and 78.93 mg g^−1^ DW of alizarin and purpurin	[41]
Stirred tank bioreactors	*Rubia tinctorum*, cell suspension culture	-	-	AQs—Not reported	[78]
New Brunswick fermenters; 1.5 and 5 L	*Rubia tinctorum*, cell suspension culture	B5 medium + 2 mg L^−1^ NAA + 0.1 mg L^−1^ IAA + 0.2 mg L^−1^ KN + 2% sucrose	The effect of hydrodynamic stress on cell viability, biomass, and anthraquinone production	AQs—70.7 μmol g^−1^ day^−1^	[79]
Stirred tank bioreactors	*Rubia tinctorum*, cell suspension culture	B5 medium + 2 mg L^−1^ NAA + 0.1 mg L^−1^ IAA + 0.2 mg L^−1^ KN + 2% sucrose	The effect of turbulence and light irradiation on cell viability, biomass, and anthraquinone production	AQs—681.3 μmol L^−1^	[80]

^1^ B5—Gamborg B5 medium; BA—benzyl adenine; IAA—indoleacetic acid; KN—kinetin; MS—Murashige and Skoog medium; NAA—1-naphthalene acetic acid.

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
