# Peer review of "Anthraquinone Production from Cell and Organ Cultures of Rubia Species: An Overview"

_metabolites, 2022, doi:10.3390/metabo13010039_

Round 1
Reviewer 1 Report
In this review, the potential of employing plant cell and tissue culture methods to increase anthraquinone synthesis is investigated. The hyperaccumulation of anthraquinones in cell and organ cultures has been investigated using a variety of techniques, including elicitor treatments and bioreactor cultures. The manuscript is well-structured and well-discussed. However, some points should be checked and corrected before it's accepted in this journal.
Therefore, according to my comments, I recommended the publication of the paper after minor revision.
[1] R. cordifolia, R. tinctorum, and R. peregrine should be in italic. In many places, genus and species did not have italics in the MS.
[2] Please add some more information about Elicitation strategies for anthraquinone production.
[3] Please add some details about Agrobacterium-mediated gentic transformation (A. tumefaciens) and Overexpression of biosynthetic genes were enhanced anthraquinone production.
[4] The MS English needs to be improved. The article's English must be carefully checked for grammatical errors.
Author Response
Ref: Metabolites-2054570
The authors are thankful to the anonymous reviewers for their valuable comments on the manuscript. We have revised the manuscript in the light of reviewers' comments and incorporated all the suggestions made by them.
Query #1. R. cordifolia ....
Answer: All the genera and species names have been shown in italics in the revised manuscript.
Query #2. Please add.......
Answer: Eliciatioan strategies used with callus, cell, and root cultures have been incorporated.
Query #3. Please as some .....
Answer: AQ biosynthetic genes such as isochorismate synthase (ICS), o-succinyl benzoate synthase, and other genes have been worked out by Shkryl et al. and this information is included in the revised manuscript.
Query # 4. The MS English.....
Answer: All the grammatical and typographical errors have been corrected.

Reviewer 2 Report
The manuscript reviews the state of the art on Anthraquinone production by in vitro plant cell culture. The paper describes the different approaches used to produce AQs by suspension cultures and hairy roots of Rubia species: Rubia tinctorum, R. cordifolia and R. akane. They also summarize the classical strategies used in plant cell culture: media optimization, phytohormone balance, elicitation and metabolic engineering. Finally, the authors describe the production of AQs by plant cell cultures in bioreactors.
The manuscript and the goal described are clear, but the paper could be improved, some of the comments are showed below.
General comments.
I consider that the main advantages of plant cell culture should be mentioned in Introduction.
In the item; 2.Biosynthesis of anthraquinones. I consider that the enzymes involved in the biosynthetic pathway should be mentioned in this item.
Tables
In general, the tables could be improved, by adding the AQs accumulation levels (mg/g Fresh weight or mg/l) reached in each culture condition tested.
The name of plant, bacteria or fungi species should be in italics (in the pdf version at least they did not appear in italics). The same corrections should be done for expressions such as in vivo or in vitro.
Specific comments
Line 58
Colorants are frequently collected from the roots of Rubia plants, where they are found as sugar derivatives called glycosides.
May be this sentence could be improved? AQs are frequently founds as glycosides in roots of Rubia plants or an alternative, for the reader is obvious that many secondary metabolites are found as glycosides. It is not necessary to explain what a glycoside is.
Line 98
The hyperaccumulation of anthraquinones in cell and organ cultures has been investigated using a variety of techniques, including elicitor treatments and bioreactor cultures.
This sentence is very similar (redundant) to the previous one, in line 93.
Line 114
It should be, chorismate instead of chorismite,
Line 115
OSB synthase then catalyzes the conversion of isochorismate into o-succinylbenzoic acid (OSB) in the presence of ketoglutarate and thiamine diphosphate (TPP).
The abbreviation OSB should appear before.
o-succinylbenzoic acid (OSB) synthase then catalyzes the conversion of isochorismate into OSB in the presence of ketoglutarate and thiamine diphosphate (TPP).
Line 119
1,4-dihydroxy-2-naphthalic acid, naphtoic acid instead of naphtalic acid
Line 122
the 2-C-methyl-D-erythritol 4 phosphates (MEP), the 2-C-methyl-D-erythritol 4 phosphate (MEP),
line 131
When compared to field cultivation, plant cell and organ cultures are appealing alternatives for the synthesis of beneficial secondary metabolites. Comparing plant cell and organ cultures to field cultivation, they offer attractive alternatives for the generation of important secondary metabolites.
I consider that the main advantages of plant cell culture should be mentioned in Introduction.
Line 149
It should be , α-ketoglutaric acid instead of -ketoglutaric acid.
line170
R. crodifolia, correct to cordifolia.
Line 188
Compared to the non-transformed culture, the rolC culture showed less resistance to the inhibitors, but the rolB culture was more resistant to the medication.
The term medication is not generally used in plant science but applies to animals or humans.
Line 230
They also demonstrated that transformed cells had ROS increase brought on by light, salt, heat, and cold stress reduced.
This sentence is not clear and incomprehensive, and should be rewritten. The verb brought is not used correctly.
Line 270
In R. cordifolia cells that express the rolB gene, Bulgakov et al. [28] showed that this gene reduces the generation of reactive oxygen species (ROS) brought on by paraquat, menadione, and light exposure.
This sentence is not clear and incomprehensive, and should be rewritten. The verb brought is not used correctly.
Line 281
R. cardifolia , correct to cordifolia.
Line 307
0, 10, 20, 30, 40, 50, 150, and 250 mg L-1), inorganic ions (KNO3, 0.1, 0.2, 0.3, 0.4, 0.5, 0.6, 0.7, 0.8, 0.9, and 1.0%), nitrogen source (NH4 or NO3) or combination of NH4:NO3, 1:3, 1:2, 1.1, 2.1, 3:1), auxins (NAA, IAA, 2,4-D; 0.2, 1.0 mg L-1), NAA (0.0, 0.2, 0.4, 0.65, 0.8, 1.0, 2.0, 3.0, 4.0, 5.0 mg L-.
I think it is not necessary to mention all concentrations tested, using a range might be better.
Line 317
Fluorescent light was advised for R. cordifolia cell culture since pH changes did not affect the biomass or AQ accumulation…
The verb advised is not used correctly, so the sentences remains unclear.
Line 374
Their findings showed that hairy roots evoked with 150 mg L-1 chitosan elicited a 10-fold greater concentration of alizarin in comparison to control cultures.
This sentence is not clear and incomprehensive, and should be rewritten. The verb evoked is not used correctly.
In table 5
R. cardifolia , correct to cordifolia.
Author Response
Ref: Metabolites:
Query #1. I consider ....
Answer: Advantages of plant cell culture for the produciton of secondary metabolites have been incorporated in the introduction.
Query #2. Biosynthesis ....
Answer: Key enzymes involved in the biosynthesis of anthraquinones have been incorporated in the figure as well as the text.
Query #3. Tables ......
Answer: The tables have been revised as per the suggestions and AQ content (yield and productivity) has been mentioned.
Query #4. The name.....
Answer: The suggestions have been incorporated.
Query #5. Line 58
Answer: The suggestion has been incorporated.
Query #6. Line 98
Answer: The suggestion has been incorporated
Query #7. Line 114
Answer: The suggestion has been incorporated
Query #8. Line 115
Answer: The suggestion has been incorporated
Query #9. Line 119
Answer: The suggestion has been incorporated
Query # 10. Line 122
Answer: The suggestion has been incorporated
Query # 11. Line 131
Answer: The suggestion has been incorporated.
Query # 12. Line 149
Answer: The suggestion has been incorporated
Query # 13. Line 170
Answer: The suggestion has been incorporated.
Query # 14. Line 230
Answer: The suggestion has been incorporated.
Query # 15. Line 270
Answer: The suggestion has been incorporated.
Query # 16. Line 281
Answer: The suggestion has been incorporated.
Query # 17. Line 307
Answer: The suggestion has been incorporated.
Query # 18. Line 317
Answer: The suggestion has been incorporated.
Query # 19. Table 5
Answer: The suggestion has been incorporated.

Round 2
Reviewer 2 Report
I revised the second version a found more little mistakes:
Line 249
They also demonstrated that transformed cells had ROS increase brought on by light, salt, heat, and cold stress reduced.
This sentence is not clear and still incomprehensive, and should be rewritten.
Line 361
-ketoglutarate through retrosynthesis.
Please correct α-ketoglutarate
Line 415
effects of various concertation of IAA, IBA, or NAA (0, 0.1, 0.5, and 1 mg L−1) added..
correct to concentration
Line 423
The effects of 2% levels of fructose, galactose, glucose, lactose, maltose, and sucrose as well as changed nitrogen (1.90 g L−1 KNO3 + 1.65 g L−1 NH4NO3 and 6.07 g L−1 KNO3) added to MS medium on the synthesis of AQ were investigated by Kino-oka et al.
This paragraph is difficult to follow, can the author rephrase it?
Author Response
Response to reviewer comments
Ref: Metabolites-2054570
The authors are thankful to anonymous reviewers for their valuable comments on the manuscript. We have revised the manuscript in the light of reviewer’s comments and incorporated all the suggestions made by them. Following are the specific changes made in the revised manuscript.
Reviewer # 2
Line 249 – They also demonstrated….
Answer: The sentence has been presented comprehensively.
Line 361-ketoglutarate
Answer: The word has been modified as α-ketoglutarate.
Line 451 – Effect of various……
Answer: Correction is incorporated.
Line 423. The effects of ………
Answer: The paragraph has been presented comprehensively.
